

# Formation of CuO$_2$ sublattices by suppression of interlattice correlations in tetragonal CuO

Max Bramberger[1,2,†], Benjamin Bacq-Labreuil[3,†], Martin Grundner[1,2],
Silke Biermann[3,4,5,6], Ulrich Schollwöck[1,2], Sebastian Paeckel[1,2] and Benjamin Lenz[7,*]

**1** Arnold Sommerfeld Center of Theoretical Physics, Department of Physics,
University of Munich, Theresienstrasse 37, 80333 Munich, Germany
**2** Munich Center for Quantum Science and Technology
(MCQST), Schellingstrasse 4, 80799 Munich, Germany
**3** CPHT, CNRS, Ecole Polytechnique, Institut Polytechnique
de Paris, F-91128 Palaiseau, France
**4** Collège de France, 11 place Marcelin Berthelot, 75005 Paris, France
**5** Department of Physics, Division of Mathematical Physics,
Lund University, Professorsgatan 1, 22363 Lund, Sweden
**6** European Theoretical Spectroscopy Facility, Europe
**7** Institut de Minéralogie, de Physique des Matériaux et de Cosmochimie, Sorbonne
Université, CNRS, MNHN, IRD, 4 Place Jussieu, 75252 Paris, France

† These authors contributed equally.     ⋆ benjamin.lenz@sorbonne-universite.fr

## Abstract

We investigate the tetragonal phase of the binary transition metal oxide CuO (t-CuO) within the context of cellular dynamical mean-field theory. Due to its strong antiferromagnetic correlations and simple structure, analysing the physics of t-CuO is of high interest as it may pave the way towards a more complete understanding of high-temperature superconductivity in hole-doped antiferromagnets. In this work we give a formal justification for the weak-coupling assumption that has previously been made for the interconnected sublattices within a single layer of t-CuO by studying the non-local self-energies of the system. We compute momentum-resolved spectral functions using a Matrix Product State (MPS)-based impurity solver directly on the real axis, which does not require any numerically ill-conditioned analytic continuation. The agreement with photoemission spectroscopy indicates that a single-band Hubbard model is sufficient to capture the material's low energy physics. We perform calculations on a range of different temperatures, finding two magnetic regimes, for which we identify the driving mechanism behind their respective insulating state. Finally, we show that in the hole-doped regime the sublattice structure of t-CuO has interesting consequences on the symmetry of the superconducting state.

# 1   Introduction

Despite an unprecedented research effort for the last 35 years, the nature of high-temperature superconductivity in cuprates and its proximity to other exotic phases like pseudogap and charge-density phases still remain elusive [1–6]. In the quest for a microscopic theory for the

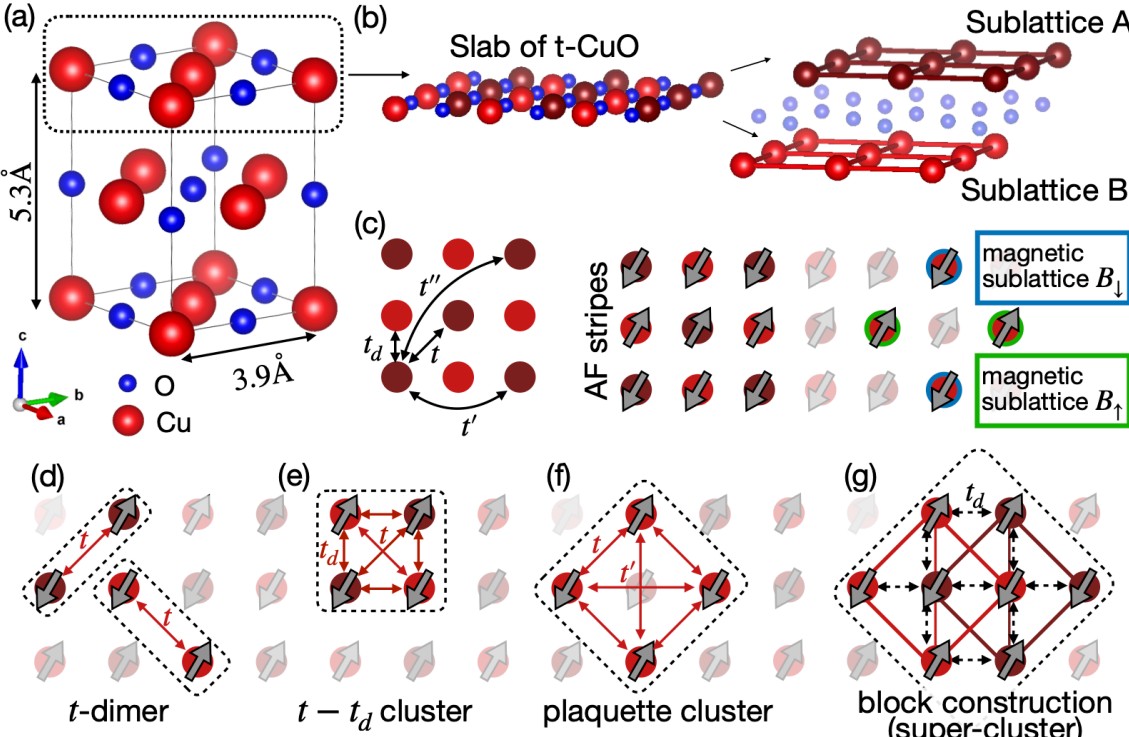

Figure 1: (a) Rock salt crystal structure of tetragonal CuO. (b) slab of CuO within the a-b plane. Bright (dark) red atoms indicate the sublattice A and B of our model. (c) Two identical Cu-sublattices and indication of the hoppings $t_d$, $t$, $t'$ and $t''$ included in the model. The arrows sketch the columnar magnetic order corresponding to an ordering vector $Q = (0, \pi)$ considered throughout the paper. Highlighted in blue and green are the magnetic sublattices that correspond to this ordering. (d-g) Clusters including different hopping terms as discussed in the text.

cuprates' superconductivity, their $CuO_2$ planes were early on identified to be key and quasi-two-dimensional (2D) minimal low-energy models were proposed and studied [7–12]. In order to connect model calculations with real materials, an *ideal* cuprate without any ligand field, distortion or disorder effects was long sought after, and polymorphs of pure Cu-O planes suggested themselves [13]. However, in contrast to other binary transition metal oxides (MnO, FeO, CoO, NiO) CuO does not crystallize in a cubic or tetragonal phase that is made up of CuO planes. Instead, a lower-symmetry monoclinic structure is realized [14].

This changes when thin films of CuO are grown on a $SrTiO_3$ substrate: CuO then crystallizes in a tetragonal crystal structure, which is composed of 2D CuO planes that are arranged in a staggered configuration along the *c*-axis [15–17]. In its distorted rocksalt structure, shown in Fig. 1(a), the Cu-O distances for basal and apical oxygens differ by a factor 1.37 [15,16].

First principles studies including density functional theory (DFT) with hybrid functionals [18–21] and DFT+U [22,23] gave first insights into the electronic structure of tetragonal CuO (t-CuO) and were able to reproduce the experimentally observed tetragonal distortion [19], which could be traced back to Jahn-Teller orbital ordering at the Cu $d^9$ ions [18,21].

*Ab initio* calculations also proposed a columnar magnetic order in (1,0) [or (0,1)] direction in units of our lattice model [18,19,21], which is in agreement with experimental findings from resonant inelastic x-ray scattering (RIXS) [24]. Extrapolation from other binary transition metal oxides [15,25] and estimates from first principles calculations [18,19,21] place the Néel temperature around $\sim 800$K, which is much higher than the critical temperature of its monoclinic bulk phase ($T_N \sim 220$K [26]). It is due to these observations that we will also study

magnetic properties within the framework of our quantum cluster methods choosing clusters that allow for a columnar magnetic order with ordering vector $Q = (0, \pi)$. In the following we will refer to this ordering as magnetic stripe order. Please note that within this paper we did not study charge order as we it is not expected to occur at half-filling, in particular since non-local interactions were not taken into account.

t-CuO is an insulator with quite sizeable gap $\Delta > 2.35\,\text{eV}$ of which the electronic structure was measured via angle-resolved photoemission spectroscopy (ARPES) [17] and used to construct effective three- and one-band $t - J$ models [17,27,28]. Whereas the question whether or not a Zhang-Rice singlet (ZRS) [10] band can describe the low-energy spectral features of t-CuO [27,28] is a (re-)current question in cuprate materials [29,30], the effective one-band model derived from RIXS in Ref. [24] is in qualitative agreement with the one derived from a ZRS description [28].
ARPES measurements [17] show strong replica features outside the single sublattice (see Fig. 1(b)) Brillouin zone (BZ), corroborated by RIXS [24] measurements of the t-CuO magnon dispersion that exhibits a strong similarity to previous experimental findings for the magnon dispersion of $Sr_2CuO_2Cl_2$. This has been interpreted as a signature of weak coupling between the two $CuO_2$ sublattices and raises the question of the microscopic origin of this sublattice decoupling.

In this paper, we investigate the dynamical influence of the inter-sublattice hopping $t_d$ by the means of cellular dynamical mean field theory (CDMFT) [31–34] and motivate an efficient block-construction scheme for our cluster calculations. Our key finding is that the inter-sublattice correlations are heavily suppressed as compared to local and short-range intra-sublattice correlations, which formally justifies to regard t-CuO as weakly-coupled interlaced $CuO_2$ lattices. Using a matrix product state [35,36] (MPS)-based impurity solver working directly on the real axis [37–41] and at effectively zero temperature we can reproduce equal energy maps and momentum resolved spectral functions in remarkable agreement with ARPES measurements without the need for analytic continuation. Furthermore, we analyse the magnetic ordering in t-CuO as a function of temperature and identify two driving mechanisms for the insulating phase. Finally, we predict the presence of superconductivity (SC) upon hole-doping by applying a complementary cluster technique, the variational cluster approximation (VCA) [42]. As a direct consequence of the sublattice decoupling, we find coexistence of magnetic stripe order and superconductivity of $d_{xy}$-symmetry, whereas the usual cuprate $d_{x^2-y^2}$ order is strongly suppressed.

## 2 Model Hamiltonian

Each CuO plane of t-CuO is made up of edge-sharing $CuO_4$ plaquettes, which can be viewed as consisting of two interpenetrating $CuO_2$ square lattices. Following this logic, we consider one slab within the $a - b$ plane as shown in Fig. 1(a). We consider a single-band Hubbard model [43]:

$$H = U \sum_i n_{i\uparrow} n_{i\downarrow} + \sum_{\substack{i,j,\sigma \\ |\mathbf{i}-\mathbf{j}|=a}} t_d c_{i\sigma}^\dagger c_{j\sigma} + \sum_{\substack{i,j,\sigma \\ |\mathbf{i}-\mathbf{j}|=\sqrt{2}a}} t c_{i\sigma}^\dagger c_{j\sigma} + \sum_{\substack{i,j,\sigma \\ |\mathbf{i}-\mathbf{j}|=2a}} t' c_{i\sigma}^\dagger c_{j\sigma} + \sum_{\substack{i,j,\sigma \\ |\mathbf{i}-\mathbf{j}|=2\sqrt{2}a}} t'' c_{i\sigma}^\dagger c_{j\sigma} ,$$

with $i, j$ being site indices and $\sigma \in \{\uparrow, \downarrow\}$.
The single particle terms ($t_d = -0.1\,\text{eV}$, $t = 0.44\,\text{eV}$, $t' = -0.2\,\text{eV}$, $t'' = 0.075\,\text{eV}$) were obtained as a result of fitting the magnon dispersion, measured by RIXS, with a t-J model in Ref. [24]. Contrary to the usual $CuO_2$ planes found in cuprate superconductors, the interstitial

O atoms within one slab favor next-nearest neighbour (NNN) hopping $t$ between Cu sites rather than nearest-neighbour (NN) hopping $t_d$.

We use an Hubbard interaction strength of $U = 7$ eV, a significantly higher value than the one from Ref. [24] but necessary for obtaining a gap that is larger than the experimental lower bound 2.35 eV [17]. Similar values of $U$ have been used in LDA+U calculations [22, 23].

## 3 Results

### 3.1 Sublattice decoupling

At the single particle level it is hard to argue for the decoupling of the two sublattices since the nearest-neighbour hopping $t_d$ is of the same order of magnitude as the next-nearest neighbour hopping ($t_d \sim -\frac{t}{4}$). Therefore it is important to also take into account the self-energy which captures the modification of the non-interacting Hamiltonian due to the presence of electronic interactions in the correlated material.

Within the framework of CDMFT [31–34, 44–46], which has shown to be extremely insightful in the context of cuprates [32, 47–54], local interactions, hopping terms on the given cluster and dynamical fluctuations to an electronic reservoir are taken into account exactly, while longer-ranged exchange with the rest of the lattice is included on the single-particle level and enters via the self-consistency loop [55]. In CDMFT, the cluster self-energy $\Sigma(\omega)$ is a matrix-valued quantity in terms of combined cluster site-spin indices. It links the non-interacting and interacting cluster Green's functions, $\mathbf{G}_0(\mathbf{k}, \omega)$ and $\mathbf{G}(\mathbf{k}, \omega)$, via the Dyson equation

$$\Sigma(\omega) = \mathbf{G}_0(\mathbf{k}, \omega)^{-1} - \mathbf{G}(\mathbf{k}, \omega)^{-1},$$

Besides the local component, $\Sigma_{\text{loc}}(\omega)$, non-local self-energies within the cluster are accessible, which we denote with respect to the hopping term connecting the corresponding sites, e.g. $\Sigma_t(\omega)$, $\Sigma_{t_d}(\omega)$. We compute the self-energy on different impurity cluster geometries (Fig. 1(d)-(g)) and probe its influence on the coupling between the two sublattices.

To this aim, we choose the dimer cluster including the next-nearest neighbour hopping $t$ (Fig. 1 (d)) and the plaquette cluster containing two such dimers connected by the next-neighbour hopping $t_d$ (Fig. 1 (e)). The following results have been obtained by a MPS-based impurity solver [56–58] working on the imaginary axis and were computed using CDMFT at effectively zero temperature ($T = 0$ K). More details on the solver can be found in Sec. 4 and App. A.

In Fig. 2 we show selected elements of the self-energy computed for those two clusters. As shown in Fig. 2(a,b) the elements of the self-energy already included in the dimer cluster do essentially not change by considering the cluster containing a dimer on each sublattice. Indeed, the self-energy element corresponding to the inter-sublattice hopping (Fig. 2(c)) is found to be about three orders of magnitude smaller than the intra-sublattice element (Fig. 2(b)). On the other hand, the inter-sublattice hopping ($t_d$) is roughly about one fourth of the leading order hopping ($|t_d| \approx |\frac{t}{4}|$). Therefore, the inter-sublattice self-energy suppression is far from trivial and indicates that electronic correlation effects strongly favour the hopping between sites that are connected by $t$, i.e. that are part of one sublattice.

We believe that the driving mechanism behind the formation of sublattices is that the hopping elements $t_d, t, t', t''$ are not decreasing monotonically with distance. The leading order hopping is largely favoured by electronic correlations irrespective of whether it is the nearest neighbour or any higher-ranged hopping. Furthermore, in systems where hopping terms monotonically decrease with distance, the sites are all connected to each other through processes including

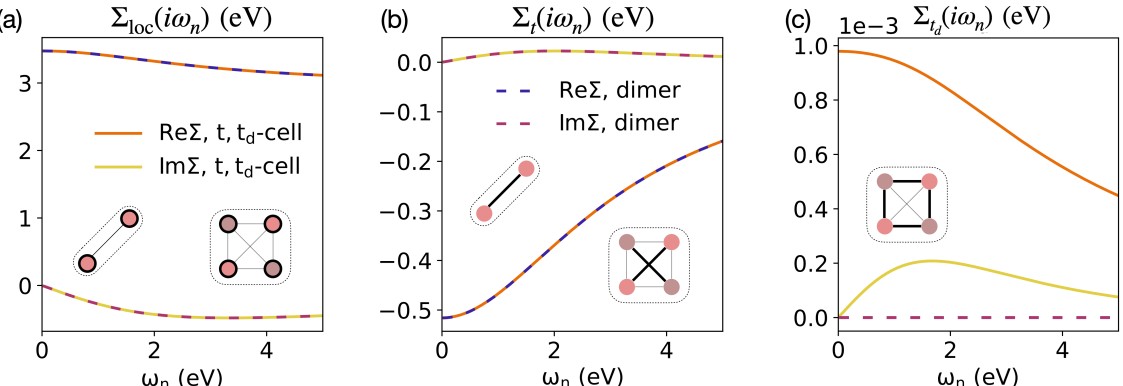

Figure 2: Comparison between selected elements of the self-energy computed on two different clusters using the MPS-based solver on the imaginary axis. Note the difference in scales between panels (a,b) and (c). The components shown belong to the block of the up-spin self-energy.

only the favoured hopping term. This leads to self-energies which smoothly decay with distance since higher-order hopping processes of the largest hopping term still connect to every site. However, due to the position of the oxygen atoms in t-CuO, the NNN hopping term ($t$) is favoured. Since the latter connects only sites from the same sublattice, the inter-sublattice self-energy shows a strong suppression. Please note that this decoupling behaviour can also be observed at finite temperatures as shown in App. B. We want to stress the importance of this result, as it proves that thinking of t-CuO as two weakly-coupled sublattices is well justified and reveals the physical origin of this behaviour.

This insight can be used to motivate a self-consistent super-cluster construction (Fig. 1(g)) consisting of two intercalated four-site intra-sublattice clusters (Fig. 1(f)) allowing us to increase the momentum resolution within our CDMFT calculations to one corresponding to an eight-site diamond cluster, while retaining the computational effort of a four-site plaquette. This super-cluster is of special interest since it allows to treat $t$ and $t'$ exactly while $t_d$ is treated perturbatively. It is moreover based on the $2 \times 2$ plaquette on each sublattice, which is argued as being the minimal cluster incorporating key ingredients of the low-energy physics of cuprates [32, 59–65]. Hereafter, we refer to the emerging cluster as block construction. Technical details of the construction can be found in App. C.

## 3.2 Spectral function

In the following, we compare calculated spectral functions using the block-construction to ARPES data. The results presented in this section were obtained by the MPS - based impurity solver on the real axis [37–41], see Sec. 4 and App. A. While the single-band Hubbard model solved with quantum cluster methods has been shown to capture the main characteristic spectral features of undoped and doped cuprates [47, 48, 66, 67], we apply this method for the first time to t-CuO.

In Fig. 3(a) we show an equal energy cut on the top of the valence band, which agrees well with the energy map measured in experiment (Ref. [17], Fig. 1(a)): We recover the strong maxima in the middle of the BZ, which are offset by 90°. We also reproduce the replica features outside the single-sublattice BZ (dashed black line) that experimentally justified the assumption of only weakly-coupled sublattices. To elaborate on this point in more detail, we note that on the one hand the two sublattices would be entirely decoupled only for $t_d = 0$, yielding the spectral function of a single sublattice. In such a case, the features inside the first

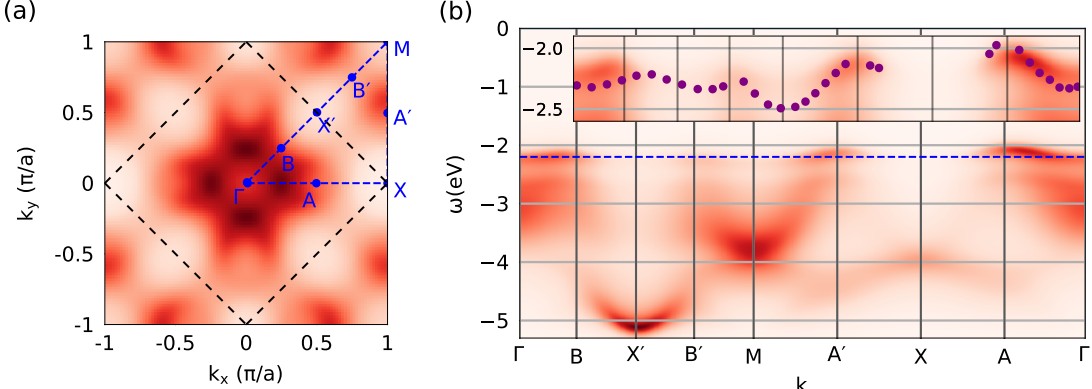

Figure 3: Spectral function $A(\mathbf{k}, \omega)$. (a) Equal energy map at $E = -2.2$ eV where the dashed black line depicts the first BZ of a single sublattice. (b) $A(\mathbf{k}, \omega)$ along high-symmetry $\mathbf{k}$-path as computed with the block-construction scheme and compared to the experimentally measured dispersion (purple circles in inset) extracted from Ref. [17] and shifted by 0.4 eV in order to align the chemical potentials. All heat maps are normalized to the maximal value displayed and averaged over the possible orientations in the block-construction (see App. D).

BZ of a single sublattice would be periodically replicated outside the BZ. On the other hand, the vanishing inter-sublattice self-energy (see Fig. 2(c)) keeps the hopping $t_d$ bare, whereas the intra-sublattice self-energy enhances the hopping $t$ (see Fig. 2(b)) by a factor of $\sim 2$. This effectively renders the $t$ hopping $\sim 10$ times stronger than $t_d$, explaining the close resemblance of the replica features with respect to the ones in the original BZ. Note that unlike in ARPES [68] there are no matrix-element effects present in our calculation, which is why our replicas do not undergo any additional intensity modulations. In order not to favor any direction by using an asymmetric super-cluster we average over possible cluster orientations. This procedure is described in more detail in App. D. The remaining difference between the $x$ and $y$ direction in Fig. 3 is entirely due to the magnetic stripe order.

In panel (b), we show the momentum-resolved spectral function of the valence band using the block construction and compare it to the ARPES spectrum along the high symmetry path through the BZ depicted in Fig. 3(a). Comparing our results to ARPES (cf. Fig. 2(a) in Ref. [17]) we find overall good agreement. In particular the low-energetic Zhang-Rice-like bands, which are separated from the lower Hubbard band at higher binding energy coincide (see inset of Fig. 3(b)). We identify this band to stem from a spin-polaron, i.e. a hole propagating in an antiferromagnetic background, similarly to the interpretation of Refs. [67,69–71] for standard cuprates. The incoherent and very dispersive spectrum without well-defined structures around the $M$ and $\Gamma$ points are also consistent with the measured spectrum. Moreover we reproduce the experimentally observed missing spectral weight at the $X$ point, a feature which was not obtained within a self-consistent Born approximation calculation based on a Zhang-Rice singlet (ZRS) [10] spin-model [28]. An obvious feature that the calculations presented in this work can not reproduce are the contributions from a lower-lying band marked with $\beta$ in the experimental data [17], which is not included in our low-energy model. However, apart from these features the agreement between our model and the experiment is striking.

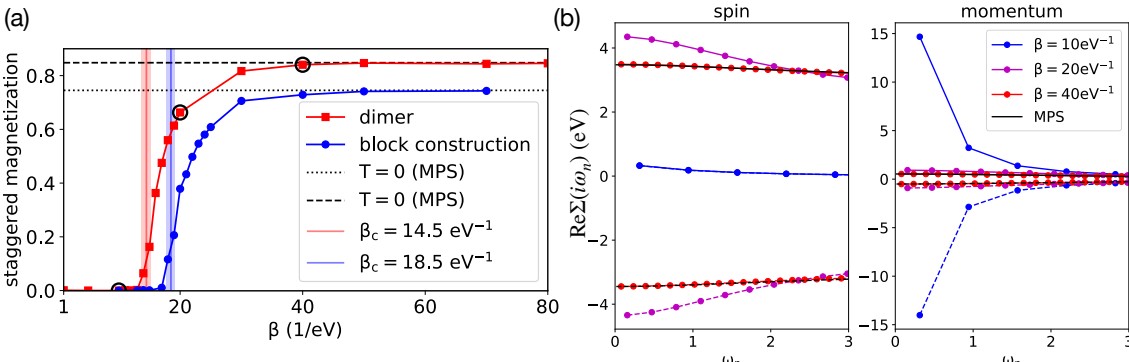

Figure 4: (a) Staggered magnetization calculated using the dimer cluster and the block construction. The dashed black lines indicate the $\beta = \infty$ result computed with the MPS based impurity solver on the imaginary axis. The vertical lines depict the inverse critical temperature $\beta_c = (18.5 \pm 0.7)\,\text{eV}^{-1}$ $((14.5 \pm 0.8)\,\text{eV}^{-1})$ for the block construction (dimer) cluster. The shaded area depicts the error bar for $\beta_c$. (b) Real part of the diagonal components of the self-energy for different inverse temperatures $\beta$ indicated in (a). The curves shown left correspond to the spin up (solid) and down (dashed) components on a cluster site. On the right, we show the self-energy at the two cluster momenta $K_1 = (0,0)$ (dashed) and $K_2 = \left(0, \frac{\pi}{a}\right)$ (solid) respectively.

## 3.3 Finite temperature analysis

All results so far presented were computed at $T = 0\,\text{K}$, however, there have been multiple predictions about the Néel temperature $T_N$ for the antiferromagnetic ordering of t-CuO in the literature [15, 18, 19, 25], which underlines the necessity to better understand the finite-temperature behavior of the system. To this end, we employ CDMFT with a continuous-time Quantum Monte Carlo solver using the dimer and the block construction clusters (Figs. 1(d),(g)).

In Fig. 4 we show the staggered magnetization as a function of temperature as well as the spin- and momentum-resolved self-energy for three characteristic temperatures.

First, we note asymptotic convergence of the staggered magnetization towards the $T = 0\,\text{K}$ value obtained with the MPS - based solver for $\beta \to \infty$. Most importantly, Fig. 4(a) allows to identify an inverse temperature at which the order melts, namely $\beta_c \approx 18.5\,\text{eV}^{-1}\,(14.5\,\text{eV}^{-1})$, corresponding to a critical temperature of $T_c \approx 627\,\text{K}\,(800\,\text{K})$ obtained with the block construction (dimer) cluster. Details about the estimation of $T_c$ can be found in App. F.

While the dimer cluster overestimates magnetic order, the block construction, which includes slightly longer-ranged magnetic fluctuations, leads to a smaller value of $T_c$. We study a simplified 2D model of t-CuO, which does not include the inter-layer magnetic exchange coupling. Long-range AF magnetic order should hence not be stable at finite temperature due to fluctuations between the two equivalent stripe configurations [72]. In fact the staggered magnetization of our CDMFT calculations is rather a consequence of choosing one of the two possible stripe directions within the mean-field scheme, than an actual hallmark of long-range magnetic order. Despite prohibiting a direct determination of $T_N$, the reduction of $T_c$ upon extending the cluster size nevertheless shows the importance of including in-plane spin fluctuations.

In Fig. 4(b), the Matsubara self-energies of the dimer cluster are compared at $T = 0\,\text{K}$ and at three characteristic temperatures corresponding to the paramagnetic (PM), the magnetically ordered and the transition region of the phase diagram.

First, as the system enters the insulating ordered phase, we observe the asymptotic conver-

gence towards the MPS (T=0) result (see Fig. 4(b)). The frequency dependence of the self-energy gets strongly suppressed. This is well described in the atomic limit as derived in Ref. [73], or by the asymptotic development of the self-energy which becomes static in the antiferromagnetic ordered limit [74].

We note that even at $\beta = 10\,\text{eV}^{-1}$ the material is still insulating as the diagonal of the Green's function (not shown) still approaches 0 in the limit of $\omega_n \to 0$. In the PM phase this can not be attributed to a freezing of dynamics due to large spin polarization, but by a momentum-selective level splitting (right panel of Fig. 4(b)): Close to the real axis, the $K_1 = (0,0)$ orbital is very strongly favoured with respect to the $K_2 = (0, \frac{\pi}{a})$ orbital. This is consistent with previous quantum cluster calculations performed for dimer and larger clusters [75, 76], and can be interpreted as a freezing of the electron movement that is not generated by spin polarization but rather by penalizing electrons with non-zero momentum.

Overall, this underlines that there is correlation-driven static level splitting present in the ordered phase whereas the PM phase is driven by dynamic splitting of momentum orbitals. We note in passing the enhancement of Re$\Sigma$ at $\beta = 20\text{eV}^{-1}$ (c.f. Fig. 4(b)) near $\omega_n = 0$. Even though here the system already is in the ordered phase, the splitting is larger than $U$ at low frequencies. This can be traced back to thermal fluctuations (see App. H for a more detailed discussion).

### 3.4 Superconductivity away from half-filling

In order to address the question of superconductivity upon doping the system, we employed the variational cluster approximation (VCA) method [42, 77, 78]. This technique is particularly well suited to study the energetics of different symmetry-breaking solutions of the model and their competition. It is based on finding stationary points of the self-energy functional $\Omega(\Sigma)$, which approximates the grand potential of the (lattice) system in the space of cluster self-energies [79]. These self-energies are parametrized via suitably chosen one-body parameters of the quantum cluster, potentially augmented by Weiss fields to allow for solutions with long-range order.

We checked for different singlet-pairing channels and found stable solutions for superconducting Weiss fields of $d_{xy}$ and $d_{x^2-y^2}$ symmetry. These two pairing channels have been discussed already in the context of the $t - t' - U$ Hubbard model [80], which would correspond to only take into account $t_d$ and $t$. Whereas the $d_{x^2-y^2}$ channel and its competition with Néel antiferromagnetism are important close to half-filling for $t/t_d < 1$, the $(\pi, 0)$ collinear antiferromagnet and SC of $d_{xy}$ symmetry were identified to be key for $t > t_d$ [80, 81]. Here, however, we focus on the superconducting channels of $d_{xy}$ and $d_{x^2-y^2}$ symmetry *away* from half-filling.

For both $d_{xy}$ and $d_{x^2-y^2}$ symmetry, the superconducting solutions are energetically favoured over the normal state (PM) solution for fillings $n < 1$. The same is true for antiferromagnetic stripe order (AFS), see Fig. 5, which we find even lower in energy down to $n \approx 0.87$. However, when allowing for competition between these symmetry-breaking orders, a coexistence of superconductivity and AFS (AFS+SC) leads to the overall lowest energy solution at zero temperature, see red curve in Fig. 5.

Comparing the corresponding antiferromagnetic and superconducting order parameters of these solutions shows that they are reduced in the coexistence solution as compared to the pure AFS and SC solutions. This indicates a competition between magnetic and superconducting orders upon doping. Most interestingly, the order parameter of the $d_{x^2-y^2}$ SC is strongly suppressed by the presence of antiferromagnetic stripe order such that the Cooper pairing is mainly of $d_{xy}$-symmetry.

Finally, we note that SC of $d_{xy}$ symmetry actually corresponds to $d_{x^2-y^2}$ symmetry within each of the two sublattices. Therefore, in the context of sublattice decoupling, our energetically

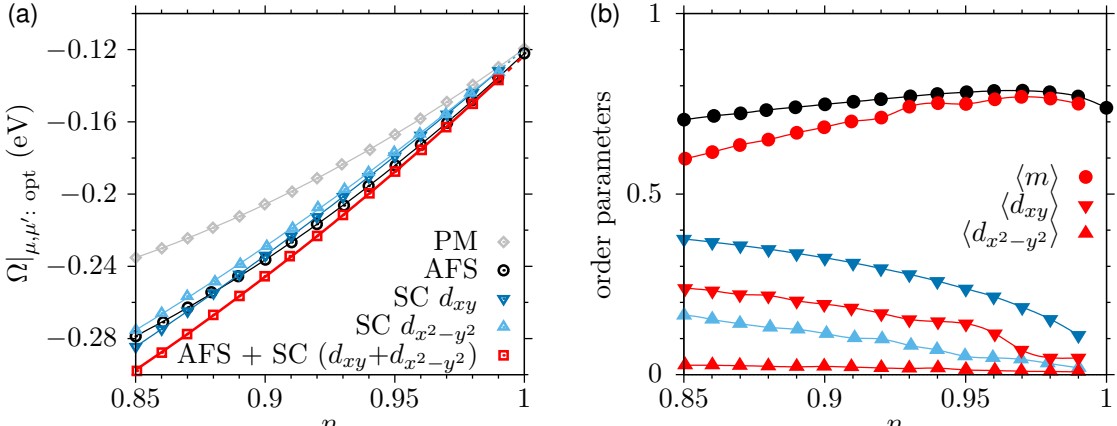

Figure 5: (a) Internal energy $\Omega$ as a function of filling $n$ for different solutions within VCA: Antiferromagnetic stripe order (AFS), superconductivity (SC) of $d_{xy}$ or $d_{x^2-y^2}$ symmetry as well as coexistence of all three. (b) Order parameters corresponding to the phases in (a); the colors correspond to the solutions in (a). For all calculations we used the full 8-site diamond cluster of Fig. 1(g), i.e. without block construction and optimized the functional in addition with respect to the (cluster) chemical potential $\mu$ ($\mu'$).

most favourable solution could be interpreted as the emergence of a $d_{x^2-y^2}$ superconducting state with coexisting (Néel) antiferromagnetic order on each sublattice.

# 4 Methods

Our method of choice to treat the interacting many-electron problem is a cluster extension of dynamical mean-field theory (CDMFT) [31–34, 82]. In CDMFT, the full lattice problem is mapped to an effective cluster of several sites which is dynamically coupled to an electronic reservoir that represents the rest of the solid. This cluster-bath system is solved numerically for its Green's function and linked to the full lattice problem via a self-consistency loop.

Due to the long-range magnetic stripe order, see Fig. 1(c), we perform magnetic calculations choosing cluster tilings, that are inline with the order. We propose several cluster geometries as shown in Fig. 1(d)-(g) for our CDMFT calculations, both to investigate the question of weak coupling (Fig. 1(d,e)) as well as to obtain further observables like the spectral function (Fig. 1(d,f,g)). In order to allow for a polarized solution, the CDMFT loop was initialized with a strongly polarized (constant) self-energy.

To investigate possible superconducting solutions upon doping, we use the variational cluster approximation (VCA) [42, 77, 78], which is an established variational quantum cluster technique well suited to check for different symmetry-breaking orders of the lattice system [83, 84]. Both techniques can be explained in terms of self energy functional theory [79, 85] and are in this sense complementary [86]. They rely on the solution of the embedded cluster problem, for which we used different solvers as detailed below.

## 4.1 Imaginary axis MPS impurity solver

We use the MPS-based impurity solver introduced in Ref. [56] and successfully applied in the context of DFT+DMFT in Refs. [57,58,87]. Using Matrix Product States (MPS) [35,36] we are able to access effectively zero temperature. MPS need a Hamiltonian formulation of the cluster impurity problem, which we obtain by following the fitting procedure introduced in Ref. [88] in the context of exact diagonalisation (ED). However with MPS we are able to treat a larger amount of bath sites allowing for a very accurate description of the hybridisation function. In this work we use $L_b = 8$ ($L_b = 6$) bath sites per spin and site yielding a total system size of $L_{\text{tot}} = 36$ ($L_{\text{tot}} = 56$) for clusters including two (four) sites.

As soon as the parameters for the impurity model are obtained we perform a grand canonical ground state search by searching every symmetry sector in the vicinity of the total particle number $N_{U=0}$ the impurity model would have in the absence of electron-electron interaction. We subsequently add single particle (hole) excitations onto the impurity sites and perform imaginary time evolution until the excitations are decayed. The interacting impurity Green's function is computed by evaluating the overlaps with the initial states. Finally a Fourier transform allows us to obtain the interacting impurity Green's function on the imaginary frequency axis, which can be used to close the self-consistency loop.

More details can be found and a comparison with CTQMC can be found in App. A and App. G respectively.

## 4.2 Real axis MPS impurity solver

The MPS based solver can also be applied on the real axis directly [37–39], allowing one to access real frequency data without the need of analytic continuation, which is known to be numerically ill-conditioned [89–91].

This allows for a good resolution on the entire frequency range. However the price to pay is that in order to discretise the hybridisation function at low broadening one needs to include a vast number of bath degrees of freedom. The discretisation procedure we use is the linear discretisation approach introduced in Ref. [92]. In this work we use a broadening of $\eta = 0.05$ eV which we treat using $L_b = 274$ ($L_b = 200$) bath sites per spin and impurity yielding a total system size of $L_{\text{tot}} = 1100$ ($L_{\text{tot}} = 1608$) for calculations with clusters including two (four) sites.

Again we search for the ground state in a grand canonical manner in the vicinity of the total particle number of the non-interacting problem $N_{U=0}$. The convergence of the ground state search is aided by preparing an initial state that is, up to truncation, the ground state of the non-interacting problem.

Once we have obtained the ground state we add single particle (hole) excitations on the impurity sites and perform real time evolutions to obtain the retarded Green's function. In contrast to imaginary axis calculations, where entanglement stays roughly constant throughout time evolutions, it grows in real axis calculations.

In order to keep this growth in check we split the time-evolution in two parts, namely a forward and backward evolution, so that the Green's function can be obtained by computing the overlap between the two for a given excitation.

Finally we Fourier transform the retarded Green's function and obtain the real frequency Green's function $G(\omega + i\eta)$ at some broadening $\eta$. Similar tensor network based real frequency single-site DMFT calculations have already been presented in Refs. [37, 38, 40, 41].

## 4.3 CTQMC

CDMFT calculations at finite temperatures were performed using the hybridization-expansion-based CT-HYB [93] solver, based on CTQMC [94] method and ALPSCore libraries [95].

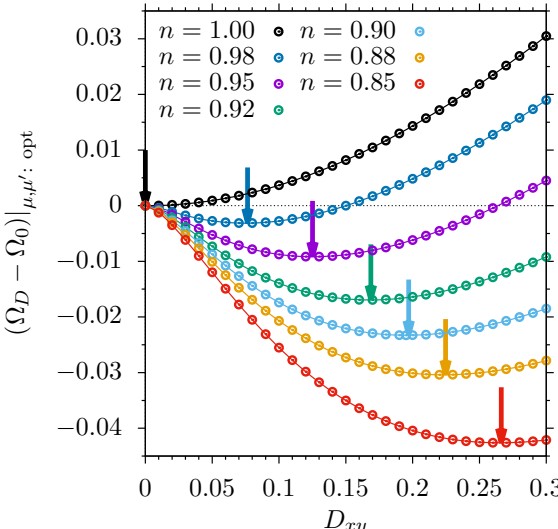

Figure 6: Self-energy functional as a function of the superconducting Weiss field strength of $d_{xy}$ symmetry, $D_{xy}$, for different electron filling $n$ at the optimal value of the chemical potential of the cluster $\mu'$. The minima of the functional are indicated by arrows. The value at zero field-strength, $\Omega_0$, has been subtracted for convenience.

Apart from the initialization, no symmetry-breaking was enforced during the self-consistency. We however explicitly used the real-space symmetries of the 2x2 plaquette. A good fermionic sign was ensured by a rotation to a basis diagonalizing the on-site energy matrix when solving the effective impurity model.

## 4.4 VCA

Variational cluster approximation (VCA) is a quantum cluster technique based on the determination of stationary points of the self-energy functional [42, 79, 85]. The search of these stationary points is limited to cluster self-energies, which are parameterized by a small number of suitably chosen cluster one-body parameters.

Here, we ensure thermodynamically consistent filling $n$ by including the chemical potential of the cluster in the set of variational parameters [96]. Furthermore, we use a Legendre-transform of the self-energy functional to specify a target filling $n$ [97]. Thereby, the chemical potential of the lattice system represents a second variational parameter. Finally, different symmetry-breaking Weiss fields were added to the cluster Hamiltonian to allow for long-range order [83, 84, 96].

To allow for antiferromagnetic stripe order with ordering wavevector $\mathbf{Q} = (0, \pi)$, we add a suitable field term on the cluster

$$\mathcal{H}_{\text{AFS}}^{\text{Weiss}} = M \sum_{\mathbf{R}} (-1)^{R_y \cdot \pi} \left( n_{\mathbf{R}\uparrow} - n_{\mathbf{R}\downarrow} \right),$$

where we denote the cluster sites with $\mathbf{R}$; $M$ is the field strength, determined via the variational principle. Likewise, superconducting pairing fields are added via

$$\mathcal{H}_{\text{SC}}^{\text{Weiss}} = D \sum_{i,j} \Delta_{i,j} \left( c_{i\uparrow} c_{j\downarrow} + \text{h.c.} \right),$$

where $D$ denotes the variational parameter and $\Delta_{i,j}$ is chosen such that it amounts to pairing with $d_{xy}$- or $d_{x^2-y^2}$-symmetry.

To calculate the cluster self-energy we use exact diagonalization with a band Lanczos algorithm [98]. Moreover, we employ a Nambu transformation to include the SC fields [84]. The existence of broken-symmetry solutions manifests in the form of additional stationary points of the self-energy functional. For instance, Fig. 6 shows the self-energy functional as a function of the field strength of the superconducting Weiss field of $d_{xy}$ symmetry. The stationary point at $D = 0$ corresponds to the normal state solution, whereas away from half-filling the minimum for $D \neq 0$ amounts to the superconducting solution, which is lower in energy. To determine the stationary points of the coexistence solution, we add successively additional Weiss fields by adiabatic switching on the field while keeping $\Omega$ stationary with respect to the other variational parameters.

More information about technical details of the calculations can be found in App. I.

## 5 Conclusion

In this paper we analyse the spectral properties of t-CuO by using a square lattice model motivated by ARPES and RIXS measurements [17, 24] that consists of two interlaced $CuO_2$ sublattices coupled by an inter-sublattice hopping $t_d$. By considering selected elements of Matsubara self-energies we show that the inter-sublattice hopping has very weak influence on dynamic correlations. Thereby, we give a formal justification for the description of t-CuO by two weakly interconnected sublattices [24] which explains the weak symmetry breaking found in ARPES [17]. In addition we present momentum-resolved spectral functions and equal energy maps computed within CDMFT directly on the real axis [37–41] using an MPS-based impurity solver and compare them to data obtained from ARPES [17], which yields good agreement. We perform calculations at finite temperatures with which we identify the driving mechanism for the insulating states found in the ordered as well as the PM phase.

Given the good agreement of our results with experiment, we believe that a minimal one-band Hubbard model is sufficient to capture most electronic and magnetic properties of t-CuO as long as dynamical local and short-range fluctuations are treated properly. Further, using VCA we are able to make predictions about the presence and symmetry of superconductivity upon hole doping. We find that the decoupling of sublattices carries through to the superconducting state coexisting with antiferromagnetic stripe order. The $d_{xy}$ symmetry of the SC order parameter can be interpreted as a superconducting state of $d_{x^2-y^2}$-type within each sublattice. Due to its tetragonal symmetry, the lack of interstitial atoms between the well separated 2D CuO layers, and the fact that the electronic properties are mainly governed by its interlaced $CuO_2$ sublattices, we believe that t-CuO may be the ideal material to gain a more complete understanding of the physics behind cuprate superconductivity. Although doping of t-CuO by chemical substitution is probably not feasible experimentally, the study of doped t-CuO by other experimental techniques like space charge doping [99], which has been successfully applied to other cuprates [100], could be an option. Another interesting route to pursue experimentally consists in growing CuO layers on top of a different substrate. Recently, copper-oxide films have been grown on $Bi_2Sr_2CaCu_2O_{8+\delta}$ (BSCO), which resulted in nodeless pairing in the superconducting state [101]. Whereas the monolayer was most likely of $CuO_2$ structure, the possibility of CuO could not be ruled out and is supported by *ab initio* calculations [102]. Several layers of t-CuO grown on top of a cuprate substrate would raise the question on the node-structure of the superconducting phase again and might even allow to tune the symmetry of the SC state as a function of the number of CuO layers.

# Acknowledgments

We thank Frédéric Mila and Michele Casula for fruitful discussions. We thank the CPHT computer support team. S.P. and M.B. thank Sorbonne University for hospitality.

**Author contributions**   B.L. and S.P. conceived and managed the project following the suggestion of the material by S.B.. M.B. and U.S. developed the imaginary-time MPS solver and M.G. the real-time MPS solver. The CDMFT algorithm was implemented by M.B. for the MPS solvers and by B.B.-L. for the QMC solver and the CDMFT calculations have been performed by M.B., M.G. and B.B.-L.. The VCA code was implemented by B.L. who also performed the calculations. M.B., B.B.-L. and B.L. prepared a first draft of the paper. All authors analyzed the data, discussed the results, and commented on the manuscript. M.B. and B.B.-L. contributed equally to this work.

**Funding information**   M.B., M.G., S.P and U.S. acknowledge support by the Deutsche Forschungsgemeinschaft (DFG, German Research Foundation) under Germany's Excellence Strategy-426 EXC-2111-390814868 and by Research Unit FOR 1807 under Project No. 207383564. S.B. was supported by the European Research Council (Project No. 617196 CORRELMAT). U.S. and B.L. thank BayFrance for funding through a mobility allowance (project FK27-2019). M.B. and M.G. acknowledge funding through the ExQM graduate school and B.B.-L. through the Institut Polytechnique de Paris. M.B., M.G., S.P and U.S. acknowledge support from the Munich Center for Quantum Science and Technology. We acknowledge supercomputing time at IDRIS-GENCI Orsay (Project No. t2022091393) and at TGCC-GENCI (Project No. A0110912043).

# A   Details on Cellular Dynamical Mean-Field Theory (CDMFT) calculations

In the main text we present results for CDMFT at finite temperature computed with Continous-Time Quantum Monte Carlo (CTQMC) [94] and at effectively zero temperature obtained using a matrix product state (MPS)-based impurity solver [37–39, 56–58] both on the real and the imaginary axis.

All real frequency quantities are directly computed on the real axis using the real frequency MPS-based solver [37–39]. When computing momentum resolved spectral functions we apply the reperiodisation procedure for cluster Green's functions proposed by Sénéchal *et al.* [103–105].

## A.1   Matrix Product State (MPS)-based solver

All MPS calculations are performed using a tensor-network impurity solver [37–39, 56, 57] based on the SYTEN toolkit [106, 107].

For both real and imaginary axis computations we use two $U(1)$ symmetries namely conservation of spin in $z$-direction and particle number.

In the case of the $t$-dimer and the $t$-$t_d$-cell (cf. Fig. 1) the up and down spin sectors are degenerate up to a permutation of sites, which is why we only compute the time-evolution for one of those sectors and determine the other one from symmetry.

### A.1.1  Imaginary axis

For imaginary axis calculations we use a frequency grid corresponding to Matsubara frequencies at a (fictitious) inverse temperature $\beta_{\text{fict}} = 200 \, \text{eV}^{-1}$. In ground state searches we allow for bond dimensions of up to 2048. For the time-evolution we use the time-dependent-variational-principle (TDVP) [108,109] up to $\tau = 200 \, \text{eV}^{-1}$. Due to being in an insulating regime for most calculations the excitations decay very quickly which is why we abort the time evolution early if the norm becomes smaller than $10^{-8}$.

Over the course of the last two iterations the self-energies presented in Fig. 2 did not change by more than $7 \cdot 10^{-4}$eV on the diagonals (Fig. 2 (a)), by not more than $5 \cdot 10^{-4}$eV on the offdiagonal corresponding to the intra-sublattice hopping $t$ (Fig. 2 (b)), and by not more than $6 \cdot 10^{-6}$eV on the inter-sublattice component (Fig. 2 (c)).

To improve the numerical accuracy for the computation of the high Matsubara frequency part (tail) of self-energies we use the additional correlator introduced by Bulla *et al.* [110].

### A.1.2  Real axis

For real axis calculations we use a broadening of $\eta = 0.05 \, \text{eV}$. In ground state searches we allow for a maximal bond dimension of 1536. We time-evolve using TDVP [108,109] up until $T_{\max} = 40 \, \text{eV}^{-1}$ ($T_{\max} = 60 \, \text{eV}^{-1}$) for the block construction (dimer) calculations. Even though more costly in real time calculations we also use the additional correlator introduced by Bulla *et al.* [110] to improve the quality of the self-energy as compared to Dyson's equation. For calculations with the dimer cluster we use a mixing factor of 0.7 in the last few iterations.

### A.2  Continuous-Time Quantum Monte Carlo (CTQMC) solver

CDMFT calculations at finite temperatures were performed using the hybridization-expansion-based CT-HYB [93] solver, based on CTQMC [94] method and ALPSCore librairies [95].

We used $N_\omega = 500$ Matsubara frequencies and a grid of $N_\tau = 2001$ imaginary time points for all $\beta$, adapting the number of Legendre polynomials to the different $\beta$ values. For all $\beta$ and cluster sizes the fermionic sign was always larger than 0.7, maximal sampling count for the last iterations was larger than $7 \cdot 10^6$ for the dimer clusters, and $2 \cdot 10^6$ for the block construction. We considered the CDMFT loop converged when the change in local Green's function became smaller than $10^{-3}$.

## B  Sublattice decoupling at finite temperatures

In Fig. A1 we show the temperature dependence of the ratio between the self-energy element $\Sigma_{t_d}(i\omega_n)$ corresponding to the inter-sublattice hopping $t_d$ and the self-energy elements on the same sublattice $\Sigma_{\text{loc}}(i\omega_n)$ and $\Sigma_t(i\omega_n)$. The self-energies in Fig. A1 were obtained by CDMFT calculations on the cluster consisting out of a dimer on every sublattice (Fig. 1 (e)) exactly as in Sec. 3.1. Similar to Sec. 3.3 we employed a continous-time Quantum Monte Carlo solver for the finite-temperature calculations.

Over the entire temperature range the inter-sublattice component of the self-energy $\Sigma_{t_d}(i\omega_n)$ stays very small compared to the components contained on a single sublattice. Note that the ratios range from roughly $10^{-2}$ to less than $10^{-3}$ indicating that the sublattices seem to decouple over the entire temperature range. This observation justifies the use of the block construction scheme (cf. App. C) mentioned in Sec. 3.1 at finite temperatures. Furthermore it indicates that the sublattice decoupling seems to be independent of the magnetic ordering, as

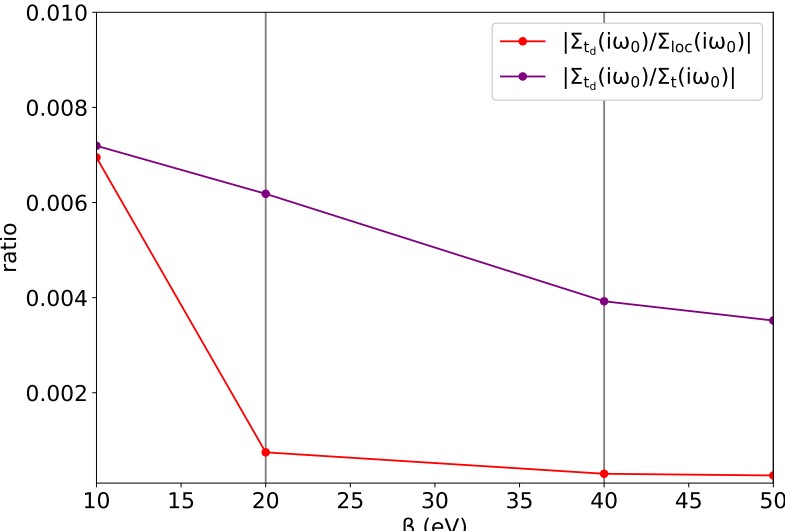

Figure A1: Temperature dependence of the ratio of $\Sigma_{t_d}$ and $\Sigma_t$, as well as the ratio between $\Sigma_{t_d}$ and $\Sigma_{\text{loc}}$. We evaluate the self energy at the lowest Matsubara frequency $\omega_0$ available for a given inverse temperature $\beta$.

even in the paramagnetic regime the inter-sublattice component is strongly suppressed.

## C  Block construction scheme

The self-consistent construction scheme consists in assuming a block structure of the cluster self-energy. One proceeds by first assuming the off-diagonals of the self-energy that correspond to $t_d$ to be zero, which is a fair approximation as we confirmed when inspecting Fig. 2. Then, the two interlaced sublattices are no longer interconnected and the cluster self-energy is of block structure when regrouping cluster site indices that belong to the same sublattice. This is for instance the case for the differently coloured sites in Fig. 1(g). Therefore, it is sufficient to solve the impurity problem on one of the two sublattices. After closing the self-consistency loop we project down onto one of those blocks and obtain an impurity problem on the unit cell of a single sublattice (Fig. 1(f) in the main text). Solving this problem yields one of the two blocks of the aforementioned self-energy.

This construction amounts to a momentum resolution as obtained by considering the cluster depicted in Fig. 1(g), but with the computational effort of considering the unit cell shown in Fig. 1(f). The approximations described in this paragraph in essence correspond to treating the inter-sublattice hopping included in the diamond on the single-particle level via a feedback from the self-consistency loop [55]. A similar block construction of a super-cluster was already used successfully within CDMFT [111, 112].

## D  Averaging dimer/diamond orientations

Most of the results in this work were computed using the $t$-dimer or the block construction as unit cells. Fig. A2(a,b) shows two different orientations for these unit cells, that are in line with the stripe order proposed in Ref. [24]. Choosing one of them would artificially introduce

an asymmetry, which is why the results presented in the paper were averaged over the two possible orientations. This approach goes by the name oriented cluster DMFT and was already introduced in Ref. [113, 114] and applied to $Sr_2IrO_4$ [113–115].

For completeness we show the equal energy maps obtained for every orientation compared to their respective mean in Fig. A2(c,d). We observe that the dimer results are far more sensitive to the orientation, however apart from the minimum in the middle of the BZ their average is already very similar to the energy maps computed with the block construction. This has two promising implications. First, it implies that the dimer results already capture very well the physics in t-CuO, indicating that the most important physical content of the extended unit cells is actually the delocalisation along the dominating bonds, as was already argued in the main text. On a second note we interpret the fact that the block construction result is almost independent of the orientation as an indication for convergence in cluster size.

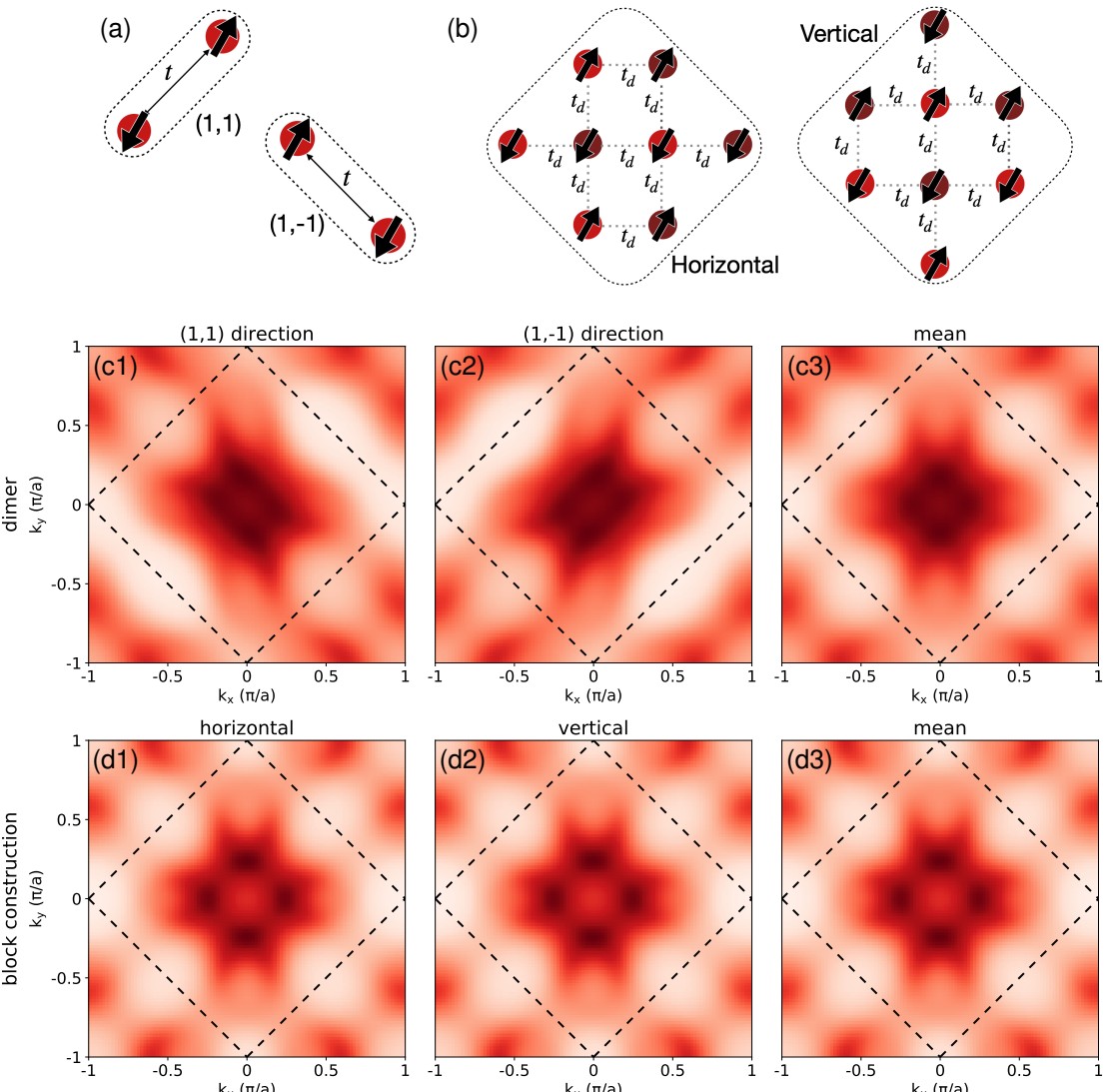

Figure A2: Sketch of the two possible orientations within a given magnetic stripe order for (a) the dimer and (b) the diamond cluster. Panels (c) and (d) show the corresponding equal energy maps obtained at $E = -2.2\,\mathrm{eV}$ using these cluster orientations as well as their mean. The dashed black line indicates the BZ of a single sublattice.

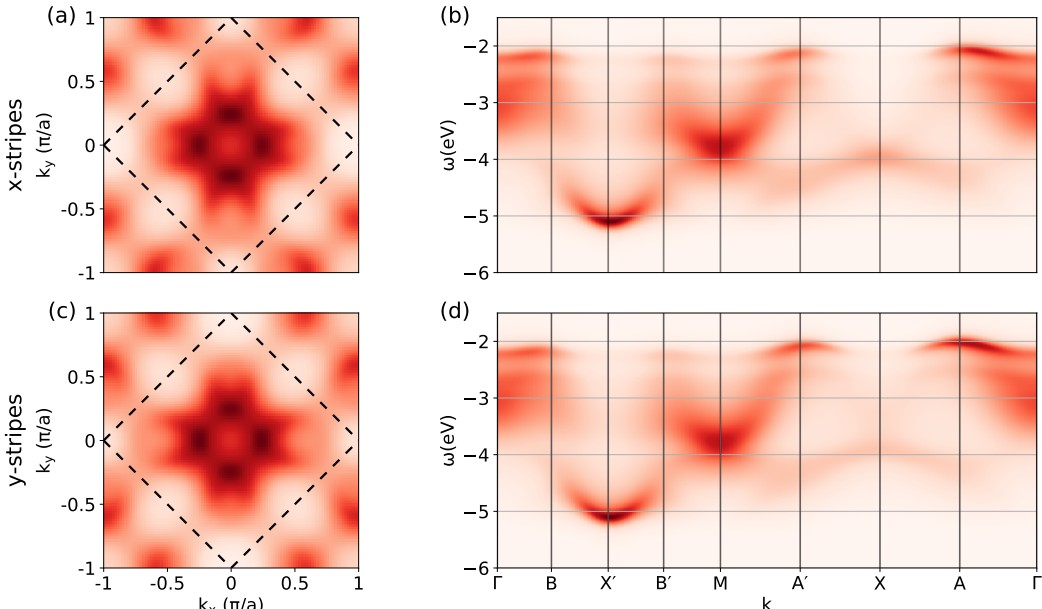

Figure A3: Comparison of equal energy maps (a,c) and spectral functions on a path through the BZ (b,d) for the two different possible directions of the stripe order. The results shown were obtained with the block construction discussed in the main text. The heat maps were normalised to the maximal value shown. The path through the BZ is the one depicted in Fig. 3(a) of the main paper.

## E  Stripe orientation

As mentioned in the main text we initialized our solvers such that the stripe order depicted in Fig. 1 is favoured. However there is no reason why the stripes should specifically be oriented in x-direction as opposed to y-direction. Thus in order to deliver a more complete picture we compare in Fig. A3 the results obtained when favouring the order in y-direction to those presented in the main text. In Ref. [27] the possibility of having multiple domains in the ARPES sample was mentioned. This would yield a spectral function that amounts to some weighted superposition of the two spectra and equal energy maps shown in Fig. A3, where the weight would depend on the portion of the sample that has a certain magnetic stripe orientation. However as both magnetic stripe orders yield qualitatively very similar results we only discuss one of them in the main text.

## F  Estimation of the critical temperature

In the main text we mention an estimate for the critical temperature and also depict it with error bars in Fig. 4(a). In order to extract this estimate we fitted a function of the form

$$M(T) = \theta(T_c - T)\gamma \left(1 - \frac{T}{T_c}\right)^{\beta},$$

to the staggered magnetization. Here $\gamma$, $T_c$ and $\beta$ are fit parameters and $\theta$ is the Heaviside step function, that was added in order to make the fits more stable. Note that unlike the rest of the manuscript here $\beta$ is the critical exponent of the transition, while $\beta_c$ in the following denotes the inverse critical temperature.

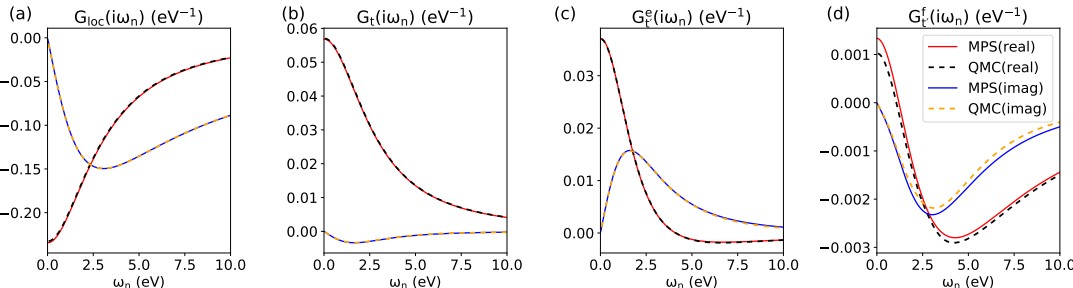

Figure A4: Real and imaginary part of chosen elements of the cluster Green's function $G(i\omega_n)$ using the horizontal block construction scheme with the MPS solver at $T = 0$ and with the CTQMC solver at $T = 1/50$ eV. Panel (a) shows $G_{\text{loc}}(i\omega_n)$ the local Green's function on a orbital that is polarized in up direction. Panel (b) shows the component corresponding to the $t$ hopping element. Panels (c,d) show components corresponding to hopping along the $t'$ direction. $G^e$ and $G^f$ in panels (c,d) stand for orbitals that are close to empty and filled respectively. Note that due to the magnetic stripe order $t'$ only connects orbitals with identical polarization (c,d), while $t$ connects those with opposite polarization (a). All the components shown correspond to the spin down block of the cluster Green's function.

By inspection of the self-energies in the transition region we find upper and lower boundaries for $\beta_c$. We set the lower boundary such that the spin splitting vanishes and the upper one such that the imaginary part of the diagonal components of the self-energy tends to 0 as $\omega_n \to 0$. By this criterion we identify $\beta_c = 16$ eV$^{-1}$ ($\beta_c = 20$ eV$^{-1}$) and $\beta_c = 13$ eV$^{-1}$ ($\beta_c = 17$ eV$^{-1}$) as upper and lower boundary for the dimer and block-construction clusters respectively.

Varying the upper and lower boundaries of the fit interval we obtain a collection of fits, of which we discard those, which either display a deviation bigger than 0.05 from any data point or which do not give $\beta_c$ in the region that was determined by inspection of the self-energies. Thus we end up with a collection of valid fits over which we average the resulting $\beta_c$. The error bars in Fig. 4(a) correspond to the standard deviation in the set of valid fits.

The average values we obtain for the critical exponent are $\beta = 0.44\pm0.15$ ($\beta = 0.66\pm0.34$) for the block-construction (dimer) respectively. The errors are again determined as the standard deviation in the set of valid fits. Finally, we note that the exponents are in good agreement with the expected mean-field critical exponent of $\beta = 0.5$ [116].

# G  Comparing CTQMC and MPS data at low temperature

At lowest temperatures (e.g. for $\beta = 50$eV$^{-1}$), the Green's functions obtained from applying the CTQMC solver and the MPS-based solver at $T = 0$ K coincide, see Fig. A4. It is important to point out that the only notable deviations occur in the off-diagonal part of the cluster Green's function, where the absolute value of $G_{i,j}(i\omega_n)$ is small ($\sim 10^{-3}$). In order to achieve good agreement between the two methods we ensured a sufficiently large sampling in the CTQMC solver: maximal sampling count was larger than $7 \cdot 10^6$ for the dimer clusters, and $2 \cdot 10^6$ for the block construction.

# H   Self-energy temperature dependence

While already being in the ordered phase, at finite temperatures (e.g. $\beta = 20\,\mathrm{eV}^{-1}$) the real part of the self-energy in Fig. 4(b) in the main text shows an additional dynamic splitting at low Matsubara frequencies. This dynamical effect decreases when temperature is lowered. The static part given by the high-frequency tail however shows a constant increase. To identify the leading mechanism we derive in the following a simple toy-model able to capture this behaviour by including thermal effects only at the single-site level.

Following the idea of Stepanov *et al.* [73], we consider a single Hubbard site subject to a small external magnetic field representing the spin-exchange coupling between neighbouring spins in a mean-field fashion:

$$H = -\mu \sum_\sigma n_\sigma - h(n_\uparrow - n_\downarrow) + U n_\uparrow n_\downarrow\,, \tag{1}$$

where $\mu = U/2$ is the chemical potential set for half-filling, $h$ is the effective field, and $U$ the on-site Coulomb interaction. The latter being larger than the other characteristic energies of the system, we assume $\beta U \gg 1$ and $\beta U \gg \beta h$. Using the finite-temperature Lehmann Green's function:

$$G_\uparrow^0(i\omega_n) = \frac{1}{i\omega_n + h + \frac{U}{2}}\,,$$

$$G_\downarrow^0(i\omega_n) = \frac{1}{i\omega_n - h + \frac{U}{2}}\,,$$

$$G_\uparrow(i\omega_n) \approx \frac{1}{(i\omega_n + h)^2 - \frac{U^2}{4}}\left(i\omega_n + h - \frac{U}{2}\tanh(\beta h)\right),$$

$$G_\downarrow(i\omega_n) \approx \frac{1}{(i\omega_n - h)^2 - \frac{U^2}{4}}\left(i\omega_n - h + \frac{U}{2}\tanh(\beta h)\right),$$

where $G^0$ and $G$ are respectively the non-interacting and interacting Green's function. The self-energy is obtained using Dyson equation:

$$\Sigma_\uparrow(i\omega_n) = i\omega_n + h + \frac{U}{2} - \frac{(i\omega_n + h)^2 - \frac{U^2}{4}}{i\omega_n + h - \frac{U}{2}\tanh(\beta h)}\,,$$

$$\Sigma_\downarrow(i\omega_n) = i\omega_n - h + \frac{U}{2} - \frac{(i\omega_n - h)^2 - \frac{U^2}{4}}{i\omega_n - h + \frac{U}{2}\tanh(\beta h)}\,.$$

First it can be checked that the derived self-energy is consistent with the CDMFT calculations in the paramagnetic $\beta h \to 0$ and the antiferromagnetic limit $\beta h \to \infty$:

$$\lim_{\beta h \to 0} \Sigma_\uparrow(i\omega_n) = \frac{U}{2} + \frac{U^2}{4(i\omega_n + h)}\,,$$

$$\lim_{\beta h \to 0} \Sigma_\downarrow(i\omega_n) = \frac{U}{2} + \frac{U^2}{4(i\omega_n - h)}\,,$$

$$\lim_{\beta h \to \infty} \Sigma_\uparrow(i\omega_n) = 0\,,$$

$$\lim_{\beta h \to \infty} \Sigma_\downarrow(i\omega_n) = U\,.$$

At high temperature ($\beta h \to 0$) we recover the Hubbard-I limit, with an additional constant $U/2$ from the chemical potential, in agreement with $\beta = 10\,\mathrm{eV}^{-1}$ data shown in the main text

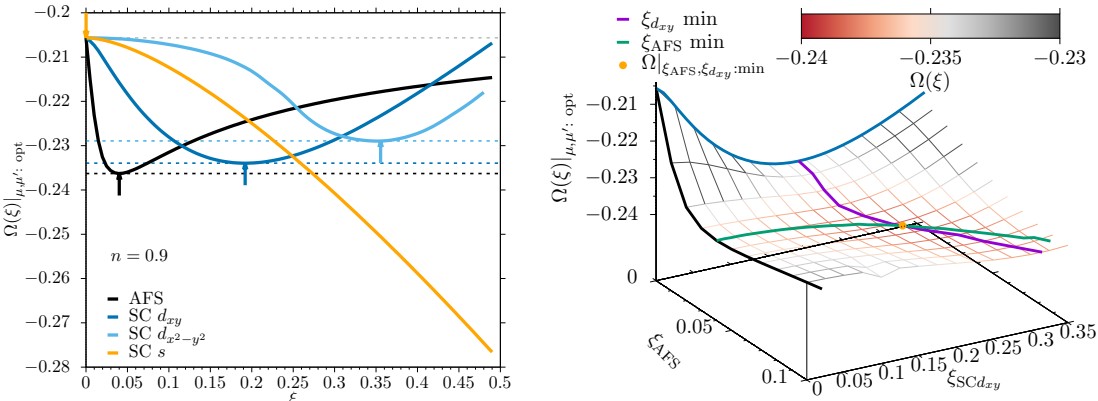

Figure A5: Left: Self-energy functional $\Omega$ as a function of different Weiss fields $\xi$ at filling $n = 0.9$. The maximum at $\xi = 0$ corresponds to the PM solution, the minima at finite $\xi$ represent solutions with broken symmetry. Right: $\Omega$ as a function of both $\xi_{\text{AFS}}$ and $\xi_{d_{xy}}$. Following the local minima of the functional along either $\xi_{\text{AFS}}$ (green) or $\xi_{d_{xy}}$ (magenta) leads to the global minimum (orange point).

which shows a vanishing real-part of the self-energy. Moreover, in the magnetically ordered limit the self-energy becomes static in agreement with CTQMC solver at $\beta = 40\,\text{eV}^{-1}$ and MPS-based impurity solver at $\beta = \infty$.

We now consider $\beta h$ finite and derive the low/high frequency limits:

$$\Sigma_{\uparrow}(i\omega_n \to \infty) = \frac{U}{2} - \frac{U}{2}\tanh(\beta h),$$

$$\Sigma_{\downarrow}(i\omega_n \to \infty) = \frac{U}{2} + \frac{U}{2}\tanh(\beta h),$$

$$\Sigma_{\uparrow}(i\omega_n \to 0) = \frac{U}{2} - \frac{U}{2}\frac{1}{\tanh(\beta h)},$$

$$\Sigma_{\downarrow}(i\omega_n \to 0) = \frac{U}{2} + \frac{U}{2}\frac{1}{\tanh(\beta h)}.$$

One can immediately see that as the temperature increases (i.e. as $\beta h$ decreases), the splitting of the tail for the two spin species decreases. However, the low-frequency limit shows an enhanced splitting larger than U and even a divergence when $\beta h \to 0$. This is perfectly consistent with the self-energy calculated with CDMFT showing at $\beta = 20\,\text{eV}^{-1}$ a larger (smaller) low (high) frequency limit than at $\beta = 40\,\text{eV}^{-1}$. Therefore we conclude that this behaviour in the ordered phase can in large parts be traced back to a pure temperature effect.

# I  Variational Cluster Approximation

The variational cluster approximation (VCA) [42] is a well-established variational method and different important technical aspects have been already discussed elsewhere [77, 78, 97].

As mentioned in the paper, we used at least the chemical potential on the cluster, $\mu'$, and the chemical potential $\mu$ as variational parameters to ensure a thermodynamically consistent filling $n$ [96, 97]. In addition, up to three additional Weiss fields, corresponding to antiferromagnetic stripe order (AFS) and superconductivity of $d_{xy}$, $d_{x^2-y^2}$ symmetry, were added. Since the search for stationary points in a high-dimensional variational space is non-trivial, we briefly outline in the following the procedure we followed.

Starting from the paramagnetic solution without Weiss field, each symmetry-breaking Weiss field $\xi_i$ was switched on individually and slowly increased. The self-energy functional $\Omega(\xi)$ was calculated for each value of $\xi_i$ while optimizing locally with respect to $\mu$ and $\mu'$. This is illustrated in the top panel of Fig. A5, where we show the form of the self-energy functional for different symmetry breaking Weiss fields. Even though the self-energy functional has a physical meaning only at its stationary points, the smooth continuous form of $\Omega(\xi_i)$ shows that adiabatically switching on the fields allows to identify new solutions while keeping the filling at a fixed value.

Once a solution with broken symmetry $\xi_1 \neq 0$ was found, we added successively additional Weiss fields by following the same procedure: Starting from a known solution, say $(\xi_1, \xi_2) = (x, 0)$ with $x \neq 0$, the field strength of the new field $\xi_2$ was slowly cranked up while assuring stationarity of $\Omega$ with respect to $\mu, \mu', \xi_1$. To cross-check solutions with more than one Weiss field having a non-zero value, we verified that the same solution was also obtained when inverting the order of successively including the Weiss fields, see right panel of Fig. A5.

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
