# Peer review of "Formation of CuO$_2$ sublattices by suppression of interlattice correlations in tetragonal CuO"

_SciPost Physics, doi:SciPost Phys. 14, 010 (2023)_

## Round 1 · Referee Report · Anonymous (Referee 1) · 2022-7-26

Report

In their manuscript ''Formation of CuO2 sublattices by suppression of interlattice correlations in tetragonal CuO" the authors investigate the tetragonal phase of the binary transition metal oxide CuO using cellular dynamical mean-field theory for finite temperatures as well as for T=0. The main result of the paper is that the inter-sublattice hopping $t_d$ can be treated perturbatively. The authors show that their results are in good agreement with ARPES experiments and that those systems can be seen as good candidates to be described by a single-band Hubbard model. The authors also investigate magnetic ordering, as well as superconducting properties in the system at finite temperatures.

The paper is clearly written, the calculations for T=0, as well as finite temperatures are well described and also comparisons between the two methods are shown. Some pertinent point, however, still needs some further clarification, before I am able to recommend the publication:

In Fig. 2 the authors show the self-energies of the onsite (a), intra-lattice hopping (b) and inter-lattice hopping (c) for zero temperature. It is well shown, that for T=0 the self-energy Sigma_{t_d} can be neglected in comparison to the onsite and intra-lattice self-energy. Since the approximation of Sigma_{t_d} being small is then used throughout the paper to create the so-called "block construction", also for finite temperature calculations, I ask the authors if they looked at the temperature dependence of
|Sigma_{loc}[n=0]| / | \Sigma_{t_d}[n=0]|
|Sigma_{t}[n=0]| / | \Sigma_{t_d}[n=0]|

Attachment

  • validity: high
  • significance: high
  • originality: ok
  • clarity: top
  • formatting: excellent
  • grammar: perfect

Author:  Max Bramberger  on 2022-09-12  [id 2806]

(in reply to Report 1 on 2022-07-26)

In their manuscript Formation of CuO2 sublattices by suppression of interlattice correlations in tetragonal CuO the authors investigate the tetragonal phase of the binary transition metal oxide CuO using cellular dynamical mean-field theory for finite temperatures as well as for T=0. The main result of the paper is that the inter-sublattice hopping $t_d$ can be treated perturbatively. The authors show that their results are in good agreement with ARPES experiments and that those systems can be seen as good candidates to be described by a single-band Hubbard model. The authors also investigate magnetic ordering, as well as superconducting properties in the system at finite temperatures.

The paper is clearly written, the calculations for $T=0$, as well as finite temperatures are well described and also comparisons between the two methods are shown. Some pertinent point, however, still needs some further clarification, before I am able to recommend the publication:

We thank the referee for reviewing our work and for the positive feedback. In the following we address the pertinent point mentioned by the referee.

In Fig. 2 the authors show the self-energies of the onsite (a), intra-lattice hopping (b) and inter-lattice hopping (c) for zero temperature. It is well shown, that for T=0 the self-energy $\Sigma_{t_d}$ can be neglected in comparison to the onsite and intra-lattice self-energy. Since the approximation of $\Sigma_{t_d}$ being small is then used throughout the paper to create the so-called block construction, also for finite temperature calculations, I ask the authors if they looked at the temperature dependence of \begin{align} |\Sigma_\textrm{loc}[n=0]| / | \Sigma_{t_d}[n=0]|\newline |\Sigma_{t}[n=0]| / | \Sigma_{t_d}[n=0]| \end{align}

We thank the referee for this important remark. We did indeed also investigate the temperature dependence of $|\Sigma_\textrm{loc}[n=0]| / | \Sigma_{t_d}[n=0]|$ and $|\Sigma_{t}[n=0]| / | \Sigma_{t_d}[n=0]|$. Our finding is that the decoupling of t-CuO into two CuO2 sublattices happens at all temperatures we studied, irrespective of magnetic ordering. We added a remark to Sec.3.1 together with a figure in the appendix showing the temperature dependence of $|\Sigma_\textrm{loc}[n=0]| / | \Sigma_{t_d}[n=0]|$ and $|\Sigma_{t}[n=0]| / | \Sigma_{t_d}[n=0]|$.

We think that our changes address the point that the referee mentioned and hope that he/she can recommend the current version of our manuscript for publication in SciPost Physics.

---

## Round 1 · Referee Report · Anonymous (Referee 2) · 2022-8-6

Strengths

State-of-the-art numerical technology

Weaknesses

Discussion of previous literature on CDMFT on copper oxide planes should be improved

Report

The manuscript investigates the physics of copper-oxide layers in the framework of dynamical mean-field theory. Due to numerical advances, the authors can achieve unprecedented precision in simulating Green's functions, which are the central object in DMFT. Thereby, the authors argue that the two sublattices in the copper-oxide planes are only weakly coupled and an effective description of the system as a single band Hubbard model is justified. The simulation clusters are coupled to a large number of bath sites, such that the simulations are far beyond the reach of Exact Diagonalization approaches. The possibility to evaluate Greens functions directly on the real axis is very exciting and an important step forward for dynamical mean-field theory. This advance is facilitated by modern and state-of-the-art matrix product state techniques for zero temperature as well as Quantum Monte Carlo simulations at finite temperature.

The manuscript is well written and also the explanations of numerical techniques is thorough, which will be of great benefit to future users of this technique. My main criticism of the manuscript is the results have not been properly put into context with previous literature. Given the wide range of literature on the topic, the novelty of all physical results obtained in the manuscript cannot fully be claimed. Therefore, The authors need to thoroughly discuss, which of the results had already been attained in previous studies and where the methodological advances surpass the literature. As presented, it seems as if all results are completely new, which is not entirely the case.

The authors mention stripe order, and sketch it in Fig. 1c). However, it is not immediately clear what they mean by this. Does stripe order refer to a combined charge and spin ordering or is it just referring to a stripy antiferromagnet, breaking lattice rotation symmetry without charge ordering? This should be explained in more detail.

Apart from these criticisms which should be addressed, I can recommend the publication of the manuscript in SciPost.
  • validity: top
  • significance: high
  • originality: good
  • clarity: top
  • formatting: excellent
  • grammar: excellent

Author:  Max Bramberger  on 2022-09-12  [id 2805]

(in reply to Report 2 on 2022-08-06)

We thank the referee for reviewing our work and for the positive feedback. In the following we will address the points of concern raised by the referee.

The manuscript is well written and also the explanations of numerical techniques is thorough, which will be of great benefit to future users of this technique. My main criticism of the manuscript is the results have not been properly put into context with previous literature. Given the wide range of literature on the topic, the novelty of all physical results obtained in the manuscript cannot fully be claimed. Therefore, The authors need to thoroughly discuss, which of the results had already been attained in previous studies and where the methodological advances surpass the literature. As presented, it seems as if all results are completely new, which is not entirely the case.

We thank the referee for pointing this out. In order to address this point, we made several corrections throughout the manuscript to better situate our work with regard to the existing literature:

  1. In Sec. 3.1, the citations referring to the framework of cluster dynamical mean-field theory (CDMFT) have been updated, and references to relevant studies using CDMFT in the context of cuprates were added. We also recall the specificity of the $2\times2$ cluster when presenting the block construction by citing a set of important studies.
  2. In Sec. 3.2, we now make clear that CDMFT has been used previously to study the spectral properties of cuprates in comparison to experiments, but that it is the first time that this technique is applied to t-CuO. Moreover, we specify that the low-energy spin-polaron interpretation is not new and we refer to relevant works on this topic.
  3. In Sec 3.3, the interpretation of the frequency suppression of the self-energy at low temperatures has been slightly modified so to show the connection to prior works. Concerning the momentum differentiation in the paramagnetic regime, references to previous studies discussing this topic using quantum cluster methods were also added.
  4. In Sec. 4, the citations referring to the framework of CDMFT have been updated.

The authors mention stripe order, and sketch it in Fig. 1c). However, it is not immediately clear what they mean by this. Does stripe order refer to a combined charge and spin ordering or is it just referring to a stripy antiferromagnet, breaking lattice rotation symmetry without charge ordering? This should be explained in more detail.

In order to make the kind of stripe order that we are addressing clearer we added a few sentences to Sec.1 stating that we study a columnar magnetic order corresponding to an ordering vector $Q=(0, \pi)$ and refer to it as magnetic stripe order. We furthermore added a similar remark to the caption of Fig.1. In addition we added a sentence clarifying that we did not study charge ordered stripes.

Apart from these criticisms which should be addressed, I can recommend the publication of the manuscript in SciPost.

We hope that our changes fully address the criticisms raised by the referee and that our manuscript can now be recommended for publication.

---

## Round 2 · Referee Report · Anonymous (Referee 1) · 2022-9-13

Report

The authors have implemented all suggested changes and addressed all of my former questions satisfactorily. Therefore I now recommend publication as a regular article in SciPost.

---

## Round 2 · Referee Report · Anonymous (Referee 2) · 2022-10-12

Report

The authors have taken into account my previous suggestions carefully. In particular, the relation to previous works has now been made more clear and, thus, I can fully recommend the manuscript for pulication.

---

## Editorial Decision

published